# *F3* Expression Drives Sensitivity to the Antibody-Drug Conjugate Tisotumab Vedotin in Glioblastoma

**DOI:** 10.3390/cancers17050834

**Published:** 2025-02-27

**Authors:** Thomas K. Sears, Wenxia Wang, Michael Drumm, Dusten Unruh, Matthew McCord, Craig Horbinski

**Affiliations:** 1Department of Neurological Surgery, Feinberg School of Medicine, Northwestern University, Chicago, IL 60611, USA; thomas.sears@northwestern.edu (T.K.S.); wenxia.wang@northwestern.edu (W.W.); michael.drumm@northwestern.edu (M.D.); 2Beckman Coulter, Chaska, MN 55318, USA; dunruh@beckman.com; 3Department of Pathology, Division of Neuropathology, Feinberg School of Medicine, Northwestern University, Chicago, IL 60611, USA; rke6tr@uvahealth.org

**Keywords:** glioblastoma, IDH mutant glioma, *F3*/tissue factor, antibody–drug conjugate, blood coagulation and hemostasis

## Abstract

Glioma is the most common type of cancer arising in the adult brain. Patients with the most deadly type of glioma, glioblastoma (GBM), have a median survival time of only 15–18 months. New treatments are therefore needed. We previously showed that gliomas with mutations in *isocitrate dehydrogenase 1* and *2* (IDH^mut^) methylate and suppress *F3*, the gene encoding Tissue Factor (TF), and that this contributes to the reduced aggressiveness of IDH^mut^ gliomas compared to high TF-expressing GBM. Recent studies showed that tisotumab vedotin, a TF antibody conjugated to monomethyl auristatin E, targets cancers with high TF levels. However, this has not yet been evaluated in gliomas. Our preclinical data indicate that tisotumab vedotin is more effective against GBM than IDH^mut^ glioma, but that intra/peritumoral hemorrhage is a side effect. This provides valuable insight into the limited therapeutic potential of tisotumab vedotin in GBM patients.

## 1. Introduction

Adult-type diffuse gliomas are an incurable form of brain cancer with a substantial societal impact; 2021 SEER data show that almost 20,000 individuals are diagnosed with glioma each year in the United States [1], and gliomas cause more average years of life lost than any other cancer [2]. Adult-type diffuse gliomas are split into three groups: (i) grade 4 *isocitrate dehydrogenase 1/2* wildtype (IDH^wt^) glioblastoma (GBM); (ii) grade 2–4 IDH1/2 mutant (IDH^mut^) astrocytoma; (iii) grade 2–3 IDH mutant (IDH^mut^) oligodendroglioma [3]. Even with current therapies (e.g., surgical resection, temozolomide, radiation, tumor-treating fields) [4], these tumors are incurable, as recurrence and progression are nearly always inevitable [5]. New treatment strategies are therefore urgently needed.

One therapeutic strategy that has yielded multiple FDA-approved drugs over the last two decades takes advantage of the antigen-specific interactions of antibodies [6]. Such antibodies usually target an extracellular region of a highly expressed membrane-bound protein on tumor cells, and may induce a therapeutic response via various mechanisms, including (i) directly blocking antigen function; (ii) triggering an antitumor immune response; and (iii) delivery of an anticancer drug that is conjugated to the antibody [7].

One membrane-bound protein that is highly expressed in a subset of gliomas is Tissue Factor (TF), which is encoded by the gene *F3* [8]. TF catalyzes the rate-limiting step in the blood coagulation cascade, but also promotes tumor malignancy via activation of EGFR, ERK, and AKT [9]. To date, the only FDA-approved drug that targets TF is an antibody–drug conjugate (ADC) used for the treatment of recurrent or metastatic cervical cancer named tisotumab vedotin-tftv (TisVed) [10,11]. This ADC is composed of a human IgG1 monoclonal antibody that binds to extracellular TF, a linker that is cleaved intracellularly, and monomethyl auristatin E (MMAE), which is delivered to the bound cell [10,12]. MMAE is a spindle poison that inhibits mitotic tubulin polymerization, and is too toxic to be used as a drug by itself [10]. In addition to delivering MMAE, the monoclonal antibody component of TisVed, tisotumab (Tis), may also inhibit tumor growth via the suppression of ERK activity [12].

Whereas IDH^wt^ GBM usually expresses high levels of TF, IDH^mut^ gliomas suppress TF expression by methylating the *F3* promoter [9,13]. This contributes to the reduced thrombogenicity and aggressiveness of IDH^mut^ gliomas [9,13]. While TisVed has been tested in a variety of carcinomas [14], it has never been tested in gliomas. GBM has also been shown by other groups to have highly prevalent and intense TF expression compared to cervical cancer, and cervical cancer is currently the only indication for which TisVed has FDA approval [10,11]. We therefore hypothesized that TisVed has therapeutic activity against high-TF-expressing IDH^wt^ GBM, but not IDH^mut^ gliomas expressing low levels of TF.

## 2. Materials and Methods

### 2.1. Cell Culture

Patient-derived glioma cultures were grown in suspension as tumorspheres in a 1:1 mixture of Neuroplex Serum-Free Neuronal Medium (GeminiBio, West Sacramento, CA, USA) and DMEM/F12 (Gibco, Waltham, MA, USA), and supplemented with Gem21 without vitamin A (1×; GeminiBio), N2 (1×; GeminiBio), EGF (20 ng/mL; PeproTech, Cranbury, NJ, USA), bFGF (20 ng/mL; PeproTech), GlutaMAX (0.5×; Gibco), heparin (0.002%; STEMCELL Technologies, Vancouver, BC, Canada), penicillin/streptomycin (1×; Gibco), and non-essential amino acids (1×; Gibco). GBM6, GBM12, GBM43, and GBM164 cells were obtained from the Mayo Clinic, whereas BT142 cells were purchased from the ATCC (Manassas, VA, USA), and TB09 cells were a generous gift by Dr. Hai Yan at Duke University. Cell lines were authenticated by short-tandem repeat and evaluated for microbiological contamination via PCR through Charles River (Wilmington, MA, USA). Mycoplasma contamination was assessed at least every 6 months via MycoStrip (InvivoGen, San Diego, CA, USA).

### 2.2. Immunoblotting

Immunoblotting was performed using Invitrogen’s Novex system according to the manufacturer’s protocol (Invitrogen, Waltham, MA, USA). Briefly, GBM12 and BT142 cell pellets were extracted with RIPA buffer using a Q800R3 sonicator (QSonica, Newtown, CT, USA), and then protein concentrations were quantified using a Bradford assay. Proteins were separated by size on a 4–12% SDS-PAGE gel for 1 h at 200 V, and then transferred for 1 h at 30 V onto a PVDF membrane. This membrane was then blocked for 30 min using Invitrogen’s StartingBlock T20 Blocking Buffer prior to primary antibody incubation overnight. The following primary antibodies were used in this study at a 1000× dilution in 5% BSA dissolved in TBST: rbGAPDH (Cell Signaling Technologies, Danvers, MA, USA; Cat# 5174) and rbTissueFactor (Novus Biologicals, Centennial, CO, USA; Cat# NBP2-67731). After overnight incubation with an anti-rabbit HRP-linked secondary antibody (Cell Signaling Technologies; Cat# 7076) diluted in StartingBlock T20, the following day, membranes were imaged on a ChemiDoc MP Imaging System (Bio-Rad, Hercules, CA, USA) using Bio-Rad’s Image Lab software V5.2.14.

### 2.3. Immunocytochemistry

Here, 8-chamber slides were coated with 0.1 mg/mL poly-L-lysine, 0.1 mg/mL poly-L-ornithine, and 1 mg/mL laminin prior to plating 5000 GBM12 cells into each chamber. After 3 days, GBM12 cells adhered to the chamber slide and were then fixed with 4% paraformaldehyde. After fixation, Tissue Factor primary antibody (Novus Biologicals; Cat # NBP2-67731) and DAPI stains were applied to the fixed GBM12 cells at a 1:100 dilution in staining buffer. For secondary antibody incubation, AlexaFluor 568 goat anti-rabbit antibodies (Invitrogen; Cat# A11011) were applied at a 1:1000 dilution in staining buffer. Images were taken on a Nikon C2 confocal microscope using the Advanced Research NIS Elements software program V5.42.03 (Nikon, Manato City, Tokyo, Japan).

### 2.4. Chemical and Biological Reagents

The following compounds and reagents were used in this study where indicated: protease-free BSA (Sigma-Aldrich, St. Louis, MO, USA; A4919), Human IgG1 (SelleckChem, Houston, TX, USA; A2051), Tis (Seagen, Bothell, WA, USA), and TisVed (Seagen).

### 2.5. Live Cell Count and Cytotoxicity Assay

Assessments of live cell counts and cytotoxicity in response to Tis and TisVed were performed using Invitrogen’s Countess II Cell Counter according to the manufacturer’s protocol. A trypan blue nuclear stain was utilized to distinguish between live and dead cells. GBM12 and BT142 cells were trypsinized into a single cell suspension, counted, and plated in non-TC treated 12-well plates at a density of 75,000 cells/well. After 24 h incubation, cells were treated with either human IgG1, Tis, or TisVed dissolved in 0.1% BSA and administered at 500× of the final well volume. After four days incubation, cells were trypsinized and assessed on the Countess for live cell counts and percent cell death. For the washout assay, cells were re-fed daily and IgG, Tis, or TisVed re-administered.

### 2.6. Intracranial and Flank Xenografts

All animal experiments were approved by Northwestern’s Institutional Animal Care and Use Committee. Intracranial (orthotopic) xenografts were established by injecting 100,000 GBM12 cells or 50,000 BT142 cells into the right striatum of 8-week-old Nod *scid* gamma (NSG) mice, both male and female. Mice were purchased from the Jackson Laboratory, Bar Harbor, ME. Mice were then monitored for moribund behavior and/or weight loss (20% of initial bodyweight) as an assessment for mortality. All surgeries were conducted in a disinfected, uncluttered area, with aseptic techniques applied. Using a 26-gauge syringe, a hole puncture was generated in the skull at 1 mm lateral and 2 mm anterior to the bregma. Then, the contents of a Hamilton syringe (2 μL of PBS with 25 × 10^3^ cells/μL) were injected in 0.5 μL increments every 10 s until 2 μL had been injected. After delivering the 2 μL, the syringe was left in place for 2 min before being slowly removed. One week later, mice were randomized and assigned to either 4 mg/kg human IgG1, Tis, or TisVed treatment groups. This dose was chosen as multiple in vivo studies have shown the maximal antitumor effect at this dose [12,15]. Mice were then monitored for tumor growth by biweekly bodyweight measurements, and a 20% reduction in bodyweight was used as the survival endpoint.

For flank xenograft experiments, GBM12 cells were suspended 1:1 in Matrigel (Corning, Corning, NY, USA) and injected subcutaneously into the right flank of Nu/J athymic nude mice at 1.5 million cells/mouse. All flank xenograft injections took place using the aseptic technique. Caliper measurements were conducted weekly, and tumor volume was calculated using the following equation: length (mm) × width^2^ (mm^2^) × 0.5. Once the average tumor volume reached 200 mm^3^, mice were assigned to each treatment group based on tumor size, so that the average tumor size was equivalent for each group. Mice were then treated with 4 mg/kg human IgG1, Tis, or TisVed once per week over 2 weeks. Mice were then further monitored for tumor growth by weekly caliper measurements. Once the control group reached an average tumor volume of 2000 mm^3^, all mice were sacrificed, and endpoint tumor mass was measured.

H&E staining was performed on intracranial xenografts by brain dissection to isolate tumor-bearing regions. Hemorrhage was semiquantiatively scored by CH while blinded to treatment group, based on the following scale: 0 = no hemorrhage; 1 = minimal microscopic hemorrhage; 2 = small microscopic hemorrhage; 3 = moderate hemorrhage coalescing into pools; 4 = grossly visible hemorrhage.

### 2.7. TF-MP Assay

To evaluate TF procoagulant activity, the cell culture supernatants from GBM6, GBM12, GBM43, BT142, TB09, and GBM164 cultures were collected under normal growth conditions. The collected supernatant was pre-cleared by centrifugation 500× *g* for 5 min at room temperature. Next, microparticles (MP) were pelleted by centrifuging 200 µL of conditioned medium at 21,000× *g* for 15 min at 4 °C. MPs were then washed with HBSA (137 mM NaCl, 5.4 mM KCl, 5.6 mM Glucose, 10 mM HEPES, 0.1% bovine serum albumin, pH 7.4). These MP suspensions were incubated with 73.2 nM FX (Enzyme Research Laboratories, South Bend, IN, USA), 2.4 nM FVIIa (Enzyme Research Laboratories), and 10 mM CaCl2 for 2 h at 37 °C, and then the reaction was halted by adding 25 mM EDTA. The FXa substrate Pefachrome FXa (Enzyme Research Laboratories) was then added to the reaction and incubated for 15 min at 37 °C. Finally, absorbance at 405 nm was recorded using Synergy 2 Multi-Mode Microplate Reader (BioTek, Winooski, VT, USA). To determine TF procoagulant activity, a standard curve with 8 points from 0–30 pg/mL was developed using re-lipidated TF (Haematologic Technologies, Essex, VT, USA).

### 2.8. TCGA PanCancer Analysis

The UCSC Xena browser was used to compare *F3* gene expression in numerous tumor types including GBM (“GBM”) and cervical cancer (“CESC”) [16]. After removing patients with “null” values relating to *F3* gene expression, the TCGA PanCancer dataset utilized for this analysis included data from 11,014 patient tumors spanning 33 different cancer types. F3 gene expression is plotted as median values with upper and lower quartiles denoted. Statistical significance was assessed using a one-way ANOVA test.

### 2.9. Statistics

Statistical analyses in this study were performed using GraphPad Prism 10.1.2 (GraphPad, Boston, MA, USA). All in vitro experiments were assessed using three biological replicates. For the statistical analysis of in vitro live cell count and cytotoxicity data, a 2-way ANOVA with Tukey’s multiple comparisons was performed. TF-MP experiments were statistically analyzed by grouping data based on IDH status and performing a two-tailed unpaired Student’s *t*-test. The survival analysis of intracranial xenografts was performed to determine statistical significance via a Mantel–Cox Log-rank test. The analysis of flank xenograft caliper measurements was performed via a two-tailed unpaired Student’s *t*-test at each timepoint, whereas tumor weights were analyzed using an unpaired Student’s *t*-test only at the endpoint. Hemorrhage data were analyzed via Fisher’s exact test.

## 3. Results

We first performed a TCGA PanCancer analysis of *F3* gene expression and show that GBM has even highest median F3 expression of any cancer type testest (Figure 1A), supporting the idea that GBM may be particularly sensitive to the anticancer effects of TisVed. To investigate the cytotoxic and antiproliferative effects of TisVed in patient-derived IDH^wt^ and IDH^mut^ glioma cells, we used trypan blue cytotoxicity assays with TF^high^ IDH^wt^ GBM12 cells and TF^low^ IDH^mut^ BT142 cultures that have been shown by our group to have high and low TF expression, respectively [9]. We focused on the TF^high^ GBM12 model since it has the highest TF expression of any cell line we tested [9]. We confirmed the differential expression of TF in these cultures via western blot, and showed that our GBM12 cultures expressed membrane-bound TF (Appendix A). We treated these cultures with 0–10,000 ng/mL of IgG isotype control, Tis, or TisVed. At four days, TisVed was cytotoxic in GBM12 cells at 10,000 ng/mL, whereas it had no activity against BT142 at any concentration (Figure 1B,C). TisVed was also more antiproliferative against GBM12 cells than BT142 cells (Figure 1D,E). Unconjugated Tis antibody had no cytotoxic effect against either cell source, and only induced an antiproliferative effect in GBM12 cells at the highest dose of 10,000 ng/mL (Figure 1B–E). We originally postulated that there may be two mechanisms for anticancer activity mediated by an anti-TF antibody drug conjugate: (1) the blocking of TF function by antibody-mediated binding; (2) the delivery of the cytotoxic cargo via TF-mediated endocytosis. Our results suggest that the anti-TF function of Tis only weakly inhibits cell proliferation, and that the observed cytotoxicity is mostly due to the delivery of MMAE to the cells.

TF is alternatively spliced into soluble and membrane-bound forms [17], and GBM12 secretes a large amount of TF into the cell culture medium (Figure 1F). We therefore hypothesized that the secreted TF may prevent TisVed from directly binding to glioma cells. Consistent with this, removing secreted TF through daily media changes enhanced both the cytotoxic and antiproliferative effects of TisVed in GBM12 cells (Figure 1B,D).

Next, we tested Tis and TisVed in mice intracranially engrafted with GBM12 or BT142 cells. Once/week IP administrations of 4 mg/kg TisVed over two weeks significantly extended the median survival of mice engrafted with GBM12 by 36% (38 days vs. 28 days IgG control, *p* = 0.006), while Tis had no significant effect (30 days vs. 28 days, *p* = 0.61) (Figure 2A). Neither Tis nor TisVed had any significant effect on survival in BT142-engrafted mice (Figure 2B). To explore TisVed’s efficacy in the absence of a blood–brain barrier, we used the same TisVed regimen in mice with flank engraftments of GBM12. In that setting, TisVed reduced tumor volume by 48% at 21 days and by 42% at 28 days post-engraftment (*p* < 0.05), whereas Tis had no significant effect on tumor size (Figure 2C). Interestingly, 12/12 mice treated with Tis and 9/12 mice treated with TisVed developed hematomas in and around the flank xenografts (Figure 3). Intracerebral tumor hemorrhage was significantly more extensive in tisotumab-treated GBM12 mice vs. IgG control (semiquantitative score 2.8 vs. 1.2), but not in TisVed-treated mice vs. control (1.7 vs. 1.2) (*p* = 0.046 by Kruskal–Wallis test) (Figure 4). Generalized brain bleeding outside of the tumor was not observed in any mice.

## 4. Discussion

We previously showed that *F3* is methylated and suppressed in IDH^mut^ gliomas, and that this contributes to the reduced malignancy and thrombogenicity of IDH^mut^ gliomas compared to IDH^wt^ GBM [13,18]. Since TF enhances GBM aggressiveness and TCGA PanCancer analysis shows that GBM has higher median F3 gene expression relative to any other tumor type (Figure 1A), we sought to explore ways of targeting it. One such therapeutic, TisVed, is already FDA-approved for metastatic and recurrent cervical cancer, but to the best of our knowledge, it has not been evaluated in preclinical glioma trials [10,12]. To explore the potential of TisVed based on TF expression, we chose to utilize GBM12 and BT142, two glioma cultures that we previously established as having high and low TF levels, respectively, in a panel of six glioma cultures [9]. Our data indicate that TisVed is active against TF^high^ IDH^wt^ GBM but not TF^low^ IDH^mut^ glioma, and that this is mostly due to the antiproliferative and cytotoxic effects of the MMAE conjugate, not the inhibition of TF signaling on the cell surface. TisVed may also be effective against other TF^high^ cancers, such as head and neck cancers, pancreatic adenocarcinoma, in addition to cervical cancer (Figure 1A) [10,12].

TF^high^ IDH^wt^ GBM releases soluble TF into the circulation [17], increasing the risk of venous thromboembolism in patients [19]. We therefore hypothesized that culture medium conditioned with soluble TF from GBM cells might prevent TisVed from reaching the cell surface, whereas TF secreted by GBM cells in vivo would be continually washed away. Our in vitro washout experiments suggest this is the case, indicating that a lack of response to TisVed in vitro does not necessarily preclude efficacy in vivo. This further reinforces the idea that, while GBM highly expresses TF, some of it is secreted into the extracellular compartment vs. the membrane-bound form required for ADC endocytosis. This paradigm could also apply to other ADC cancer therapeutics if the target antigen exists in both membrane-bound and secreted forms, and may explain why TisVed showed therapeutic potential against cervical cancer in similar flank xenograft models, considering the high membrane-bound TF expression in cervical cancer [10,11,12,20].

TisVed extended the survival of mice intracranially engrafted with IDH^wt^ GBM, suggesting that this antibody conjugate can access intracerebral gliomas. TisVed did not appear to show greater activity in the flank, suggesting that the blood–brain barrier does not substantially affect therapeutic response. One limitation of our study is that we utilized immunocompromised glioma models, and therefore did not evaluate whether TisVed may also further elicit anticancer activity through the immune system.

Both Tis and TisVed caused hematomas in and around the flank tumors. Within intracerebral tumors, Tis showed significantly increased hemorrhage compared to IgG control (Figure 4). TisVed also showed a slight increase vs. control, but that difference was not statistically significant. Together with Factor VIIa, TF activates Factor X in the extrinsic blood coagulation pathway [21]. An anti-TF antibody would therefore be expected to sequester circulating TF and increase the risk of hemorrhage. Indeed, a subset of patients treated with TisVed for cervical cancer developed hemorrhage-related adverse events [20].

## 5. Conclusions

In sum, our data suggest that TisVed is active against TF^high^ IDH^wt^ GBM, primarily because the tisotumab antibody acts as a delivery system for MMAE rather than inhibiting TF signaling. Additionally, the high TF levels in GBM do not appear to render them as sensitive to TisVed in other tumor types with comparable TF levels. Therefore, if clinical trials are to be explored in GBM, limited efficacy and hemorrhagic side effects should be considered.

## Figures and Tables

**Figure 1 cancers-17-00834-f001:**
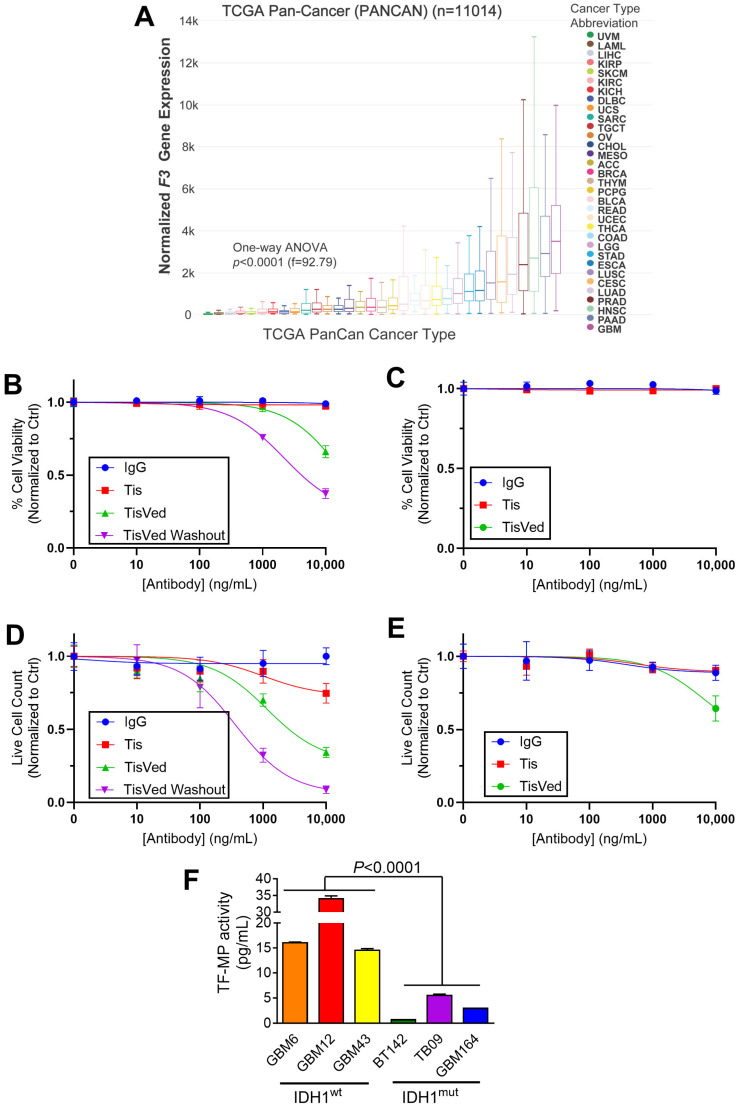
TisVed treatment, but not Tis, is cytotoxic to high-TF GBM but not low-TF IDH^mut^ glioma. (**A**) *F3* gene expression was compared between tumor types using the TCGA PanCancer dataset. (**B**–**E**) Trypan blue assays were used to evaluate antiproliferative and cytotoxic responses to TisVed. Cell death is shown for high-TF GBM12 (**B**) and low-TF BT142 (**C**), whereas live cell counts for GBM12 are seen in (**D**) and for BT142 in **E**. Daily media replacement and retreatment were performed for TisVed Washout samples in (**B**,**D**). Two-way ANOVA with multiple comparisons was used to evaluate significance at each dose *p* < 0.0001. (**F**) TF-MP activity analysis of IDH^wt^ (GBM6, GBM12, GBM43) and IDH^mut^ (BT142, TB09, GBM164) glioma cell culture supernatant. Statistical analysis was performed by Student’s *t*-test after grouping data based on IDH status. Error bars represent the SEM of three biological replicates.

**Figure 2 cancers-17-00834-f002:**
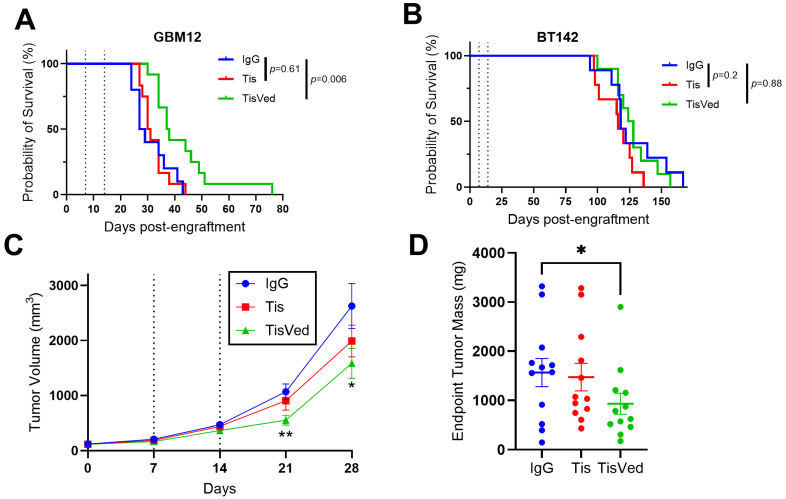
TisVed increases survival and decreases tumor growth in intracranial and flank xenograft models of GBM, but not IDH^mut^ glioma. (**A**,**B**) Intracranial xenografts were generated in NSG mice using high-TF GBM12 (**A**) or low-TF BT142 (**B**). Statistical significance was determined via Log-Rank (Mantel–Cox) Test. (**C**) Flank xenografts were generated in Nu/J nude mice using GBM12 cells where caliper measurements were recorded weekly post-tumor engraftment. Statistical analysis was performed via multiple unpaired Student’s *t*-tests. * *p* < 0.05; ** *p* < 0.01. (**D**) Flank xenograft endpoint tumor mass was assessed for each treatment group, and statistical significance was evaluated using an unpaired Student’s *t*-test. * *p* < 0.05. All intracranial and flank xenograft mice were treated with 4 mg/kg of antibody 1x/week for two weeks as denoted by the dotted lines. Error bars represent the standard deviations of 12 mice per treatment group for both intracranial and flank models.

**Figure 3 cancers-17-00834-f003:**
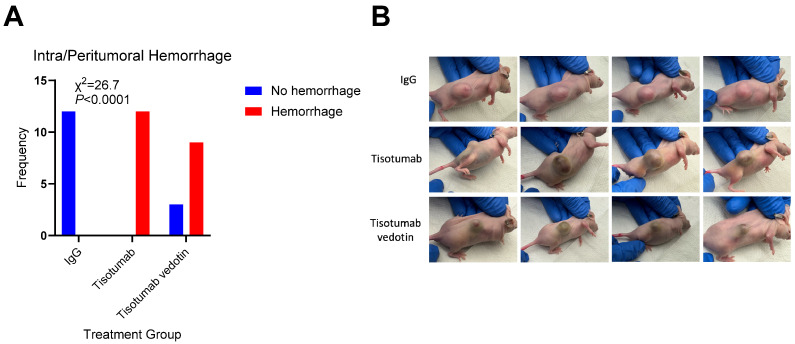
Administration of Tis and TisVed is associated with intra- and peri-tumoral hemorrhaging. (**A**,**B**) Nu/J nude mice with GBM12 flank xenografts were inspected visually for apparent hemorrhaging, and tabulation of hemorrhage is presented in (**A**) with associated images in (**B)**. Images were taken two weeks after last drug administration. Statistical significance was determined using a Fisher’s exact test.

**Figure 4 cancers-17-00834-f004:**
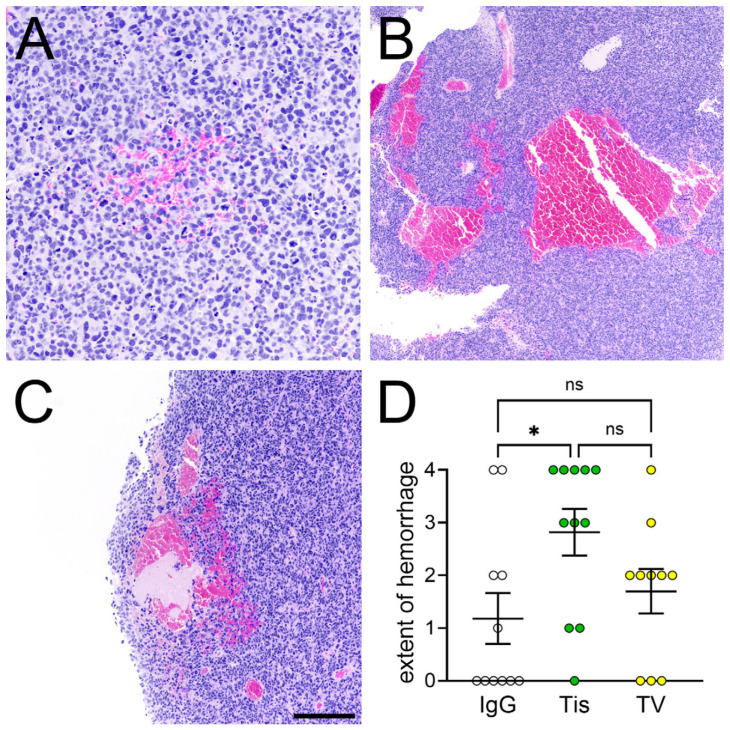
The administration of tisotumab and TV causes intracranial glioma-associated hemorrhage in NSG mice. (**A**) Photomicrograph of microscopic (score = 1) hemorrhage in a mouse engrafted with GBM12 and treated with IgG. (**B**) Large (score = 4) hemorrhage in a mouse engrafted with GBM12 and treated with tisotumab (Tis). (**C**) Moderate (score = 3) hemorrhage in a mouse engrafted with GBM12 and treated with TisVed (TV). Scale bar in (**C**) = 80 microns in (**A**), 400 microns in (**B**), and 200 microns in (**C**). (**D**) Semiquantification of glioma-associated hemorrhage in GBM12-engrafted mice according to treatment group. * *p* < 0.05 by Kruskal–Wallis test with Dunn’s multiple comparisons test. See “Methods” for scoring system. ns = not significant.

## Data Availability

All data referenced in this manuscript are available from the corresponding author upon request.

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
