# Peer review of "F3* Expression Drives Sensitivity to the Antibody-Drug Conjugate Tisotumab Vedotin in Glioblastoma"

_cancers, 2025, doi:10.3390/cancers17050834_

Round 1
Reviewer 1 Report
Comments and Suggestions for Authors
The authors focused on the high expression of F3 in GBM and conducted several experiments based on hypotheses that TisVed, which is applied to cervical cancer, is effective in GBM.
This hypothesis is a good perspective regarding the possible treatment for the condition that has not been noticed.
Here are some questions or comments below.
DAPI is used in supplementary Fig 1, but the methodology has a trypan blue explanation.
Line 141: Intracranial Xenographs
Please explain the methods and procedures of intracranial transplantation.
Line 148: For Flank Xenograph Experiments
In the Flank Xenograph Experiments, please describe that it is transplanted to the subcutaneous or muscle.
Line 191:Result
To investigate the cytotoxic and antiproliferative effects of TisVed in patient-derived IDHwt and IDHmut glioma cells, we used trypan blue assays with TFhigh IDHwt GBM12 cells and TFlow IDHmut BT142 cultures that have shown by our group to have high and low TF expression, respectively[9]. We focused on these TFhigh GBM12 model since it has the highest TF expression of any cell line we tested [9].
This explanation is the content of Reference 9 and is not done in this study. You should write the basis for using GBM12 and BT142 on Methods or Discussion.
Line 203-204: Together, these results suggest that the anti-TF function of Tis weakly inhibits cell proliferation, but that the observed cytotoxicity is due to the delivery of MMAE to the cells.
It is difficult to understand the basis of this sentence. The authors stated on line 75 as follows.
Line 75: tisotumab (Tis), may also inhibit tumor growth via suppression of ERK activity [12].
Line 212
What concentration level is the TisVed 4mg/kg dose equivalent to in the culture cell experiment? What is the basis for this dose?
Line 213: the median survival of mice engrafted with GBM12 by 36% (38 days vs. 28 days IgG control, P=0.006)
Authors should pathologically present the anti-tumor effect of TisVed for intracranial lesions. It is unclear whether the cause of death in the TisVed group is the tumor regrowth or the effect of intratumoral hemorrhage.
Line 218: Please revise "tumor volume" to "tumor area." (cf; Fig. 2C)
Line 249: TisVed did not appear to show greater activity in the flank, suggesting that the blood-brain barrier does not substantially affect therapeutic response.
I do not understand the intent of this sentence in line 249 because the sentence in Line 259: "the mass effect of flank tumors is much greater than in intracranial xenografts," is inconsistent with the sentence in Line 249.
Line 264: Additionally, the high TF levels in GBM do not appear to render them as sensitive to TisVed in other tumor types with comparable TF levels.
The basis of this sentence is unknown.
Figure 2 Legend
"Intracranial" is easier to understand than "Orthotopic" in line 377.
Author Response
Reviewer #1
Comment #1: DAPI is used in supplementary Fig 1, but the methodology has a trypan blue explanation.
Response #1: We adjusted the relevant text to more clearly state that DAPI was used in our immunocytochemistry experiments (“Immunocytochemistry” paragraph), whereas trypan blue was used for our cytotoxicity experiments.
Comment #2: Please explain the methods and procedures of intracranial transplantation.
Response #2: We now include the intracranial transplantation procedure in the “Intracranial and Flank Xenografts” section of the paper.
Comment #3: In the Flank Xenograft Experiments, please describe that it is transplanted to the subcutaneous or muscle.
Response #3: Our flank xenograft experiments included subcutaneous injections of glioma cells, not intramuscular injections. This is now clarified in the methods.
Comment #4: This explanation [TF expression in glioma cells] is the content of Reference 9 and is not done in this study. You should write the basis for using GBM12 and BT142 in Methods or Discussion.
Response #4: We now include this in the first paragraph of the Discussion.
Comment #5: Line 203-204: “Together, these results suggest that the anti-TF function of Tis weakly inhibits cell proliferation, but that the observed cytotoxicity is due to the delivery of MMAE to the cells.” It is difficult to understand the basis of this sentence. The authors stated on line 75 as follows. Line 75: tisotumab (Tis), may also inhibit tumor growth via suppression of ERK activity [12].
Comment #5: We rewrote the end of the first paragraph of the Results section as follows: “We originally postulated that there may be two mechanisms for anticancer activity mediated by an anti-TF antibody drug conjugate: (1) blocking of TF function by anti-body-mediated binding; (2) delivery of the cytotoxic cargo via TF-mediated endocytosis. Our results suggest that the anti-TF function of Tis only weakly inhibits cell proliferation, but and that the observed cytotoxicity is mostly due to the delivery of MMAE to the cells.”
Comment #6: What concentration level is the TisVed 4mg/kg dose equivalent to in the culture cell experiment? What is the basis for this dose
Response #6: The rationale for using this dose in vivo is that in a previous study by others, 4 mg/kg was shown to elicit maximal antitumor activity in animal models (see PMID: 24371232; 37828725), and as recommended by Seagen/Pfizer. These references are now included in the in vivo Methods section of the paper.
Comment #7: Authors should pathologically present the anti-tumor effect of TisVed for intracranial lesions. It is unclear whether the cause of death in the TisVed group is the tumor regrowth or the effect of intratumoral hemorrhage.
Response #7: Intracerebral tumor hemorrhage is now included as a new Figure 4. Only intratumoral hemorrhage was observed, not generalized brain bleeding.
Comment #8: Line 218: Please revise "tumor volume" to "tumor area." (cf; Fig. 2C).
Response #8: Thank you for catching this. We adjusted the y-axis in Fig 2C to represent tumor volume as we measured via caliper measurements.
Comment #9: Line 249: I do not understand the intent of this sentence in line 249 because the sentence in Line 259: "the mass effect of flank tumors is much greater than in intracranial xenografts," is inconsistent with the sentence in Line 249.
Response #9: To avoid confusion, we revised the last paragraph of the Discussion. We deleted that sentence, and added another sentence describing the new Figure 4 results.
Comment #10: “Line 264: Additionally, the high TF levels in GBM do not appear to render them as sensitive to TisVed in other tumor types with comparable TF levels.” The basis of this sentence is unknown.
Response #10: The basis for this sentence is seen in Fig. 1A where GBM has the highest median F3 gene expression out of all cancers in the TCGA-PANCAN dataset. With this in mind, previous research showed that HNSC, PAAD, and CESC tumors respond with profound tumor regression (PMID: 24371232), whereas this is not observed in our TF-high GBM model. We now explain this in the last sentence of the first paragraph of the discussion.
Comment #11: “Intracranial" is easier to understand than "Orthotopic" in line 377.
Response #11: This has been adjusted throughout the manuscript for clarity.

Reviewer 2 Report
Comments and Suggestions for Authors
This is an interesting paper about the possible therapeutic effects of Tis or TisVed on IDH wildtype GBM and IDH mutant gliomas. It is nearly impossible to find unfavorable points about the MS. I have just a few comments about it.
My comments are about the side effect of Tis or TisVed, i.e., intra/peritumoral hemorrhage. I think this can seriously preclude these agents from being one of therapeutic options. Regarding this, I want to raise the following questions. If possible, I would like the authors to answer these questions in the MS.
-Is this side effect dose-dependent? There seems to be no data about this dose-dependency in the MS. If so, is the dose of the agents, which is expected to be high enough to reach the therapeutic dose, really safely determined for them to be administered to humans?
-Reading the last paragraph of the Discussion, brain (intracranial) lesions could be safe (at least safer than flank lesions) in terms of the side effect. The authors mentioned ‘mass effect’ as a possible explanation. But I just wonder if this is really the case. Are there any data indicating the possible ‘safe or not’ about brain (intracranial) lesions if administered with Tis or TisVed?
Author Response
Reviewer #2
Comment #1: Is this side effect dose-dependent? There seems to be no data about this dose-dependency in the MS. If so, is the dose of the agents, which is expected to be high enough to reach the therapeutic dose, really safely determined for them to be administered to humans?
Response #1: The in vivo dose we used (4 mg/kg) was the dose used in a previous in vivo study by others (PMID: 24371232; 37828725), and as recommended by Seagen/Pfizer. These references are now included in the in vivo Methods section of the paper.
Comment #2: Reading the last paragraph of the Discussion, brain (intracranial) lesions could be safe (at least safer than flank lesions) in terms of the side effect. The authors mentioned ‘mass effect’ as a possible explanation. But I just wonder if this is really the case. Are there any data indicating the possible ‘safe or not’ about brain (intracranial) lesions if administered with Tis or TisVed?
Response #2: Intracerebral hemorrhage is now included as a new Figure 4. To avoid confusion, we revised the last paragraph of the Discussion. We deleted the sentence containing the term "mass effect,” and added another sentence describing the new Figure 4 results.

Round 2
Reviewer 1 Report
Comments and Suggestions for Authors
Thank you for your thoughtful response to all the comments.